# Characterization of microRNAs and Target Genes in *Musa acuminata* subsp. *burmannicoides*, var. Calcutta 4 during Interaction with *Pseudocercospora musae*

**DOI:** 10.3390/plants12071473

**Published:** 2023-03-28

**Authors:** Erica Cristina Silva Rego, Tatiana David Miranda Pinheiro, Fernando Campos de Assis Fonseca, Taísa Godoy Gomes, Erica de Castro Costa, Lucas Santos Bastos, Gabriel Sergio Costa Alves, Michelle Guitton Cotta, Edson Perito Amorim, Claudia Fortes Ferreira, Roberto Coiti Togawa, Marcos Mota Do Carmo Costa, Priscila Grynberg, Robert Neil Gerard Miller

**Affiliations:** 1Instituto de Ciências Biológicas, Universidade de Brasília, Brasília 70910-900, DF, Brazil; 2Departament of Academic Areas, Instituto Federal de Goiás (IFG), Águas Lindas 72910-733, GO, Brazil; 3Embrapa Cassava and Tropical Fruits, Cruz das Almas 44380-000, BA, Brazil; 4Embrapa Recursos Genéticos e Biotecnologia, Parque Estação Biológica, CP 02372, Brasília 70770-917, DF, Brazil

**Keywords:** *Musa acuminata*, *Pseudocercospora musae*, Sigatoka leaf spot, biotic stress, microRNAs, Illumina, stem-loop quantitative real-time PCR

## Abstract

Endogenous microRNAs (miRNAs) are small non-coding RNAs that perform post-transcriptional regulatory roles across diverse cellular processes, including defence responses to biotic stresses. *Pseudocercospora musae*, the causal agent of Sigatoka leaf spot disease in banana (*Musa* spp.), is an important fungal pathogen of the plant. Illumina HiSeq 2500 sequencing of small RNA libraries derived from leaf material in *Musa acuminata* subsp. *burmannicoides*, var. Calcutta 4 (resistant) after inoculation with fungal conidiospores and equivalent non-inoculated controls revealed 202 conserved miRNAs from 30 miR-families together with 24 predicted novel miRNAs. Conserved members included those from families miRNA156, miRNA166, miRNA171, miRNA396, miRNA167, miRNA172, miRNA160, miRNA164, miRNA168, miRNA159, miRNA169, miRNA393, miRNA535, miRNA482, miRNA2118, and miRNA397, all known to be involved in plant immune responses. Gene ontology (GO) analysis of gene targets indicated molecular activity terms related to defence responses that included nucleotide binding, oxidoreductase activity, and protein kinase activity. Biological process terms associated with defence included response to hormone and response to oxidative stress. DNA binding and transcription factor activity also indicated the involvement of miRNA target genes in the regulation of gene expression during defence responses. sRNA-seq expression data for miRNAs and RNAseq data for target genes were validated using stem-loop quantitative real-time PCR (qRT-PCR). For the 11 conserved miRNAs selected based on family abundance and known involvement in plant defence responses, the data revealed a frequent negative correlation of expression between miRNAs and target host genes. This examination provides novel information on miRNA-mediated host defence responses, applicable in genetic engineering for the control of Sigatoka leaf spot disease.

## 1. Introduction

Banana (*Musa* spp.) is a monocotyledonous crop cultivated throughout tropical and subtropical regions, contributing to food security and poverty alleviation. The fruit crop, however, is susceptible to diseases due to sterility and a narrow genetic background.

The fungal pathogen *Pseudocercospora musae,* the causal agent of Sigatoka leaf spot disease, or yellow Sigatoka disease, is a member of the Sigatoka leaf spot complex, together with the closely related species *P. fijiensis* and *P. eumusae*. Whilst *P. fijiensis* is now predominant globally, *P. musae* remains widespread and an economically important pathogen. Symptoms of this disease comprise advancing necrotic leaf spots with a reduced photosynthetic capacity leading to reduced fruit size and number [1,2]. Given the absence of genetic resistance in many commercial *Musa* cultivars, disease control is dependent upon systemic or protectant fungicides, with an associated negative impact on the environment, increased production costs, and potentially reduced sensitivity of the pathogen populations over time to applied fungicides [3].

To advance the genetic improvement of *Musa* for resistance, advances in the understanding of the molecular mechanisms controlling resistance responses are required. Considerable work has been conducted at the transcriptome level, with the unigenes potentially involved in defence responses characterized in the *Musa*-*Fusarium oxysporum* [4,5,6,7,8,9,10,11,12], *Musa*-*Meloidogyne incognita* [13] and *Musa*-*Pseudocercospora fijiensis* pathosystems [14]. Unigene sets in the *Musa*-*P. musae* pathosystem have also been characterized [15], with Illumina RNAseq recently revealing host gene expression associated with immune responses during the incompatible interaction [16].

Small RNAs (sRNAs) are short non-coding mobile RNA molecules typically 20–30 nucleotides in length. With participation in eukaryotic organisms in diverse cellular processes [17], they are classified into short interfering RNAs (siRNAs) and microRNAs (miRNAs) according to their biogenesis and function. The former class is derived from double-stranded hairpin-structured RNA transcripts, whilst the miRNA class members originate from single-stranded hairpin RNAs (hpRNA), which form stem-loop precursor secondary structures [18]. Approximately 20–24 nucleotides in length [17,19,20], miRNAs are involved in the negative regulation of protein-encoding genes at the post-transcriptional level. Complementarity with specific mRNA targets results in cleavage or, more rarely, repression of mRNA translation mediated by the miRNA-associated proteins [21,22,23,24,25,26]. Recent work has revealed transcription factors as major targets of miRNAs [27,28]. Transcribed from endogenous *MIR* genes, primary transcripts of precursor miRNAs (pre-miRNAs) form self-complementary HpRNAs 70–200 in length. With evidence that both strands of the miRNA-5p/miRNA-3p duplex are potentially active [29], these intermediates are subsequently diced by Dicer-like 1 (DCL1) endoribonucleases into short, mature miRNA double-stranded duplexes 21–24 bp in length. One of these strands will then load into Argonaute (AGO) proteins to form an RNA-Induced Silencing complex (RISC) [30]. Post-transcriptional gene silencing (PTGS) occurs via the RISC delivery of miRNA to the complementary sequence in targeted mRNA, without full complementarity. Expression repression then occurs via cleavage of the mRNAs or through the inhibition of translation [31]. As such, a single miRNA may regulate the expression of several mRNAs, and single mRNA molecules may be regulated by multiple miRNAs [32].

MiRNA families, where members are derived from a common ancestor, regulate diverse biological and physiological processes that include regulatory pathways controlling plant development, reproduction, metabolism, signal transduction, and the response to biotic and abiotic stresses [33,34,35,36,37,38,39,40]. With regard to their roles in plant immunity, miRNAs have been observed in host responses in different pathosystems. These include examples in *Solanum lycopersicum* during interaction with *Phytophthora infestans* [41]; in *Triticum aestivum* during interactions with the leaf rust pathogen *Puccinia triticina* [42] and the powdery mildew pathogen *Erysiphe graminis* f. sp. *tritici* [43]; in *Camellia sinensis* during infection by *Colletotrichum gloeosporioides* [44]; in *Oryza sativa* during interaction with *Xanthomonas oryzae* pv. *oryzae* [45]; and in *Arabidopsis* during interaction with *Phytophthora capsica* [46]. The involvement of miRNAs in specific branches of the innate immune system has also been documented. Examples include miR393, which suppresses auxin signalling and contributes to PAMP-triggered immunity (PTI) [47], as well as miR482 and miR5300, which regulate NLR gene expression in Effector-triggered immune (ETI) responses in the Solanaceae family [48,49,50,51]. Functionally validated miRNAs in the immune response to bacterial and fungal pathogens have also been described [52] (and references therein). These include members of the families miR159, miR160, miR162, miR164, miR166, miR168, miR169, miR319, miR393, miR396, miR398, miR400, miR482, miR773, miR844, miR858, miR1023, miR1507, miR1916, miR2109, miR2118, miR6019, miR6020, miR7695, and miR9863.

Genome-wide studies into *Musa* miRNAs have been conducted, with 266 miRNAs in 47 families predicted in the A genome [14], and 270 members in 42 families predicted in the B genome [53]. miRNAs have also been characterized in *Musa* under different temperature stresses [54,55,56], and in RNA materials for cultivars contrasting in resistance to *Fusarium oxysporum* f. sp. *cubense* [40]. Recently, ref. [57] characterized miR169 expression patterns in *Musa* following *F. oxysporum* f. sp. *cubense* tropical race 4 infection. Ref. [58] also reported over 110 mature miRNAs during the interaction of Cavendish banana root tissue with *F. oxysporum* f. sp. *cubense* tropical race 4 and *F. oxysporum* f. sp. *cubense* race 1.

This study was conducted to characterize miRNAs in *M. acuminata* subsp. *burmannicoides*, var. Calcutta 4 induced during the incompatible interaction with *P. musae*, together with the prediction of corresponding target plant host genes. The expression data provide insights into the miRNA-mediated defence response and represent a resource for the development of efficient methods for the control of Sigatoka leaf spot disease.

## 2. Results

### 2.1. Sequence Statistics

Illumina HiSeq 2500 sequencing of small RNA libraries was conducted on leaf material in *M. acuminata* var. Calcutta 4 harvested after inoculation with fungal conidiospores and equivalent non-inoculated controls. From a total of 255 million raw sequence reads, following adapter trimming, the removal of RNA contaminants such as tRNAs, rRNAs, and snRNAs, and the selection of only sRNA sequences between 20 and 24 nucleotides in length, a total of 25.5, 19.5, 23.7 and 23.6 million reads were retained for each of the libraries 3DAI_NI, 3DAI_I, 12DAI_NI, and 12DAI_I, respectively. Following the removal of sequences that occurred in only one of the three biological replicate libraries for each treatment, a total of 22.9, 17.8, 21.9, and 21.7 million reads were retained, representing 646.263, 378.890, 452.237, and 646.263 unique sequences for libraries 3DAI_NI, 3DAI_I, 12DAI_NI and 12DAI_I, respectively. Of these, 239.038, 136.706, 161.139, and 148.642 mapped, respectively, against the *M. acuminata* ssp. *malaccensis* var. DH Pahang reference genome.

### 2.2. Identification of miRNAs

Analysis revealed *Musa* sRNA sizes varying from 20 to 24 nt (Figure 1), with 21 and 24 nt reads representing the most abundant classes of small RNAs at the two time points, as expected for plant samples.

Following mapping, a total of 202 miRNAs were observed that belong to 30 distinct miRNA families. Of these, 166 were complete, with an miRNA:miRNA* duplex, and 36 were identified based on a mature miRNA sequence (Appendix A). A total of 24 novel miRNAs were also identified, with nine complete and 15 containing only mature miRNA sequences. Appendix A provides supporting information for all conserved and novel *Musa acuminata* var. Calcutta 4 miRNAs. Figure 2 displays examples of the structure of selected predicted novel *M. acuminata* var. Calcutta 4 miRNAs. Hairpin structures for the entire set of novel miRNAs can be found in Appendix A.

To identify the *M. acuminata* var. Calcutta 4 miRNAs potentially involved in the immune responses to *P. musae*, both the presence and expression at the two time points after inoculation were determined through examination of the sRNA-seq data.

With regard to membership across the 30 known miRNA families, miRNA156 presented the highest number of distinct miRNA gene loci, with 25 members observed across the treatments. This was followed by miRNA166 (20 loci), miRNA171 (19 loci), miRNA396 (17 loci), miRNA167 and miRNA172 (14 loci each), miR160 and miRNA164 (10 loci each), miR168 (9 loci), miR159 and miR535 (8 loci each), miR162, miR169 and miR393 (6 loci each), miR390 (5 loci), miR157 and miR529 (4 loci each), miR319 and miR395 (3 loci each), and miR391 (2 loci). All remaining miRNA families, together with each novel *Musa* miRNA, contained only one individual miRNA member (Figure 3; Appendix A).

In relation to the presence of the miRNAs across the sRNA libraries, the majority were expressed across all treatments, 3DAI_NI, 3DAI_I, 12DAI_NI and 12DAI_I (Appendix A). A comparison between 3DAI_NI and 3DAI_I revealed 221 miRNAs expressed during both treatments, with four expressed exclusively at 3DAI_NI (Novel-5p_16, Novel-3p_17, Novel-3p_19 and Novel-3p_21) and three at 3DAI_I (Novel-3p_10, Novel-5p_12 and Novel-5p_15). Similarly, at 12DAI, 216 were common to both treatments, five exclusively expressed at 12DAI_NI (three miR172a-5p members, miR168b-5p and miR169l-5p), and three at 12DAI_I (Novel-5p_12, miR169-5p and miR169I). When comparing the two time points, six miRNAs were exclusively expressed at 3DAI (Novel-5p_16, Novel-3p_21, Novel-3p_19, Novel-3p_17, Novel-3p_20 and Novel-5p_14). Figure 4 also summarizes the numbers for abundant miRNAs observed across the sRNA libraries, each with an excess of 1000 read counts. Highly abundant miRNAs comprised a total of 147 at 3DAI and 151 at 12DAI.

### 2.3. In Silico Expression Analysis of Musa miRNAs and Target Genes in Response to P. musae

Analysis of the expression changes between the treatments was conducted to identify miRNAs modulated in response to challenge by *P. musae*. *M. acuminata* var. Calcutta 4 miRNAs were identified as differentially expressed based on the mapping of reads to reference genome gene models (Appendix A), followed by an analysis using HTSeq-count and EdgeR to identify significantly differentially expressed miRNAs in inoculated treatment time point datasets versus equivalent non-inoculated controls. Relative expression of miRNAs with an FDR-adjusted *p*-value (padj) of ≤0.1 was considered significant. Across 3DAI, eight miRNAs were significantly differentially expressed, with seven up-regulated between the 3DAI_I and 3DAI_NI treatments (miR395a, miR398a, miR397a, miR827, miR408b, miR160b and miR399b) and one down-regulated (miR530a). For 12DAI, only a single miRNA was significantly differentially expressed (miR395a), with up-regulation between the inoculated and non-inoculated controls (Appendix A).

Heatmap Representation of Differential Gene Expression of miRNAs and Target Genes

A global representation of all miRNA gene expression profiles in *M. acuminata* var. Calcutta 4 during the interaction with *P. musae* is presented graphically as a global hierarchical clustering heatmap of expression patterns over time (Figure 5). Expression data, considering both the significant and non-significant expression profiles, were considered in the inoculated treatments in comparison to the non-inoculated controls at each time point. Differences in the expression profiles of miRNAs in inoculated datasets in comparison to non-inoculated datasets were apparent between the two examined time points, with similar numbers of both up- and down-regulated miRNAs observed across the entire set of miRNAs at each time point.

The in silico expression analysis of the host target genes of *M. acuminata* var. Calcutta 4 miRNA expressed during the interaction with *P. musae* is also summarized in Figure 5. Again, gene expression modulation was considered in the inoculated treatments in relation to non-inoculated controls at each time point, with red tones representing overexpressed genes and blue tones representing repressed genes. The expression data considered both significant and non-significant expression profiles, to enable a global examination of the expression tendencies. Clear differences in gene expression profiles were apparent between the two examined time points, as observed with the miRNA expression data. When comparing miRNA and target gene expression at each separate time point, the heatmaps also reveal a frequent negative correlation of the expression between miRNAs and their target genes.

### 2.4. miRNA Target Gene Prediction of Conserved and Novel M. acuminata var. Calcutta 4 miRNAs

Prediction of the target genes of mature miRNAs on the basis of complementarity using TargetFinder revealed targets for the majority of the known miRNAs and all of the putative novel miRNAs (Appendix A).

With regard to the putative novel *M. acuminata* var. Calcutta 4 miRNAs expressed in *P. musae*-inoculated and non-inoculated cDNA sets during the plant’s response to this hemibiotrophic pathogen, a diverse set of the gene targets of these miRNAs were identified. These encoded a probable mediator of RNA polymerase II transcription subunit 26b, a pollen-specific leucine-rich repeat extensin-like protein 3, an interferon-related developmental regulator 1-like, an uncharacterized protein LOC103993778, an F-box/kelch-repeat protein, a desumoylating isopeptidase 2, a transcript variant X2, a zinc finger protein ZAT11, three galacturonosyltransferase 8 proteins, a ribulose-phosphate 3-epimerase, a 60S ribosomal protein L10a-1, a putative peroxidase-25, a Homeobox-leucine zipper protein, a subtilisin-like protease SBT5.4, a glutamate receptor 3.5, a probable apyrase 7, a serine/threonine-protein phosphatase 2A 65 kDa regulatory subunit A beta isoform, a protein disulphide-isomerase LQY1, a long chain acyl-CoA synthetase 4, a G-type lectin S-receptor-like serine/threonine-protein kinase B120, a MYB1R1-like transcription factor, a putative disease resistance protein RGA1, and an ethylene-responsive transcription factor ERF054.

### 2.5. GO Analysis of Target Genes of miRNAs

Analysis of the predicted target genes of all identified known and novel *M. acuminata* var. Calcutta 4 miRNAs, according to gene ontology (GO) classifications, revealed their assignment to a total of eight cellular components, 11 biological processes, and 15 molecular function terms (Figure 6). DNA binding, protein binding and transcription factor activity (molecular activity), regulation of transcription, responses to hormones and the oxidation–reduction process (biological process), and the nucleus, ribosome and proteasome core complex (cellular components), were the most abundant categories for the putative *Musa* target genes. Molecular activity terms potentially related to defence responses to pathogen invasion included nucleic acid binding, nucleotide binding, oxidoreductase activity and protein kinase activity. Biological process terms associated with defence included response to hormone, oxidation–reduction, and response to oxidative stress. DNA binding, transcription factor activity, regulation of transcription, and the nucleus, also provided evidence for the involvement of numerous *Musa* miRNA target genes in the regulation of host gene expression during defence responses to pathogen invasion.

### 2.6. RT-qPCR Analysis of Expression of Musa miRNA and Musa Target Genes

For the validation of the expression of *M. acuminata* var. Calcutta 4 miRNAs and their target genes in the plant, stem-loop RT-qPCR and conventional RT-qPCR were conducted using the original total RNA employed for the RNAseq and small RNA sequencing, from each of the three biological replicates at 3DAI. A total of 11 miRNAs from seven families, together with their predicted target genes, were selected for an expression analysis based on miRNA family abundance and their known involvement in defence responses in plants. The miRNA–target gene combinations tested were as follows: miR397a-5p-Ma03_g12850; miR393b-3p-Ma06_g18000; miR171a-3p-Ma06_g38750; miR171e-3p-Ma06_g38750; miR156b-5p-Ma09_g28300; miR167a-3p-Ma09_g28300; miR167b-3p-Ma11_g01530; miR167a-5p-Ma11_g01530; miR167a-Ma11_g01530; miR164c-5p-Ma11_g24070; and miR535d-5p-Ma08_g07960. The specific RT-PCR primers for the miRNAs and the target genes are listed in Appendix A. RT-PCR amplification resulted in expected product sizes at 80 nt, confirming the presence of the miRNAs in the sequenced RNA (Appendix A).

Although the expression modulation observed by small RNA sequencing was generally not significant on the basis of the EdgeR analysis, with only eight miRNAs with significant differential expression in the inoculated samples in comparison to controls at 3DAI and only one at 12DAI, stem-loop RT-qPCR validation of the expression of biotic stress-induced *Musa* miRNAs, and conventional RT-qPCR validation of the target genes, revealed a general agreement with the expression tendencies observed using small RNA sequencing and RNA-seq, for most samples (Figure 7). The data also revealed miRNA and target genes displaying a negative correlation of expression in most cases, consistent with the sequencing-derived expression data.

## 3. Discussion

Although the understanding of the function of miRNAs in plant development is, to date, more consolidated, the analysis of the expression of miRNAs and their target host genes is fundamental in determining the roles of miRNAs in plant immune responses. In this evolving area, where miRNAs can function in PTI and/or ETI responses [52], numerous plant miRNAs have now been functionally characterized in plant immune responses [59,60,61,62,63,64,65]. Although *Musa* miRNAs have been characterized in relation to responses to drought, salt, and temperature stresses [56,66,67], as well as in relation to evolutionary changes across different *Musa* subgroups [68], analyses in the context of biotic stress responses have been limited to cultivars contrasting in resistance to *F. oxysporum* f. sp. *cubense* races [40,57,58]. No investigation, prior to this study, has been conducted on miRNA characterization and expression analysis in the *M. acuminata–P. musae* pathosystem. Here, analysing both inoculated plant leaf tissues and non-inoculated controls in the resistant wild diploid *M. acuminata* var. Calcutta 4 at early and later time points during the plant–pathogen interaction, a total of 202 conserved *Musa* miRNAs were identified, belonging to 30 miR-families, together with 24 predicted novel miRNAs. The numbers of miRNAs in each family varied, with the miRNA families 156, 166, 171, 396, 167, 172, 160 and 164 harbouring most members. A similar abundance for the families 156, 167 and 396 was observed in *Arabidopsis* miRNAs expressed in response to *Phytophthora capsica* [46], with [58] also observing abundance in *Musa* for the families 156, 166, 167, 171, 395 and 396 during interaction with *F. oxysporum* f. sp. *cubense* races.

### 3.1. Expression Analysis

At each time point, a comparison of miRNA expression was made between the inoculated plant leaf tissues and non-inoculated controls. Across all the treatments, members of the *Musa* miRNA families 399, 171, 398, 159, 166 and 396 displayed the highest expression levels, consistent with previous findings in *Musa* during interaction with *F. oxysporum* f. sp. *cubense* tropical race 4 and *F. oxysporum* f. sp. *cubense* race 1 [58]. At 3DAI, eight miRNAs, representing different families, were significantly differentially expressed in silico between the inoculated and control plant leaf samples. At 12DAI, by contrast, only a single miRNA was identified as differentially expressed. An accurate analysis of *M. acuminata* var. Calcutta 4 miRNA expression by RT-qPCR for 11 miRNAs and target genes was also conducted following the selection of abundant miRNA family members in the study that are known to be involved in plant defence responses. The data revealed expression tendencies generally compatible with in silico expression at 3DAI, supporting both the reliability of the sRNA-seq results and previous mRNAseq data for the target genes [16]. Evidence has been provided in recent years that many miRNAs are highly conserved across multiple plant species, cleaving identical or similar target genes. As such, miRNAs function as gene repressors, with a negative correlation of expression between miRNAs and their gene targets. The data here also revealed the up- or down-modulation of miRNAs and a general inverse expression to target host genes.

### 3.2. Abundant miRNAs and Target Genes Potentially Involved in the Plant Immune Response

Several abundant miRNA families and target genes characterized in the *M. acuminata* var. Calcutta 4–*P. musae* interaction are potentially involved in the plant immune response and are discussed below.

MicroRNA156 is a conserved plant miRNA family known to target members of the Squamosa Promoter binding protein-Like (*SPL*) family of transcription factor genes. These genes, which are associated with the regulation of plant growth phase changes, flowering, fertility, secondary metabolite production, and JA responses, have been reported in numerous plant species [69,70,71,72,73,74]. Modulated positively and negatively in diverse plant species in response to viral, fungal and nematode pathogens (for example, [43,74,75,76]), in *Musa,* this miRNA family has been characterized in Cavendish and Silk subgroups [68] and has been reported to be up-regulated in response to temperature stress [56] as well as during interaction with *F. oxysporum* f. sp. *cubense* TR4 [58]. Here, miRNA156 was the most abundant family in the study, with 25 members characterized across the treatments. In silico differential expression analysis during interaction with *P. musae,* although not significant, showed examples of both greater numbers of reads as well as fewer reads for miRNA156 members in relation to non-inoculated controls, with a generally greater abundance at 12DAI. RT-qPCR data at 3DAI for the miRNA–target gene pair miR156b-5p-Ma09_g28300 revealed a negative correlation with the expression of the target *Musa SPL* gene that suggests an involvement of this miRNA family in the regulation of the target gene in the biotic stress response to *P. musae*.

The miRNA166 family target transcription factors from the HD-ZIP homeobox family involved in plant development [77]. Negative regulation of ABA signalling by miRNA166 members may also favour SA-mediated immune pathways [78]. Fei et al. [58] reported the greatest abundance of this miRNA in *M. acuminata* Cavendish cv. ‘Williams’, with up-regulation in response to inoculation with *F. oxysporum* f. sp. *cubense* TR4. Similarly, up-regulation has been reported during the incompatible response of *Oryza sativa* to *Magnaporthe oryzae* ([52] and references therein, 2020), after infection in *Musa* with Banana Streak Mysore virus [79] as well as during powdery mildew infection in *Triticum aestivum* [43], and *Glycine max* responses to *Pseudomonas sojae* [80]. Interestingly, the latter authors also confirmed that miRNA166 is responsive to PAMPs, through investigation of heat-inactivated *P. sojae*. Here, the miRNA family was also abundant, with 20 members observed across the treatments. In silico differential expression analysis during interaction with *P. musae,* although not significant, nevertheless, indicated evidence for modulation, with increased expression in relation to controls at 12DAI, as well as down-regulation of members at both time points. Such down-regulation has also been described in the defence response to the hemibiotroph *Colletotrichum graminicola* in *Zea mays* [81].

MicroRNA171 family members have been shown to cleave mRNA products of SCARECROW-LIKE (*SCL*) transcription factor genes, which regulate root and shoot development and plant hormone signalling. Down-regulation in the *Arabidopsis* interaction with *H. schachtii* promotes cyst nematode parasitism [76]. Here, with 19 members characterized, read abundance, although not significant, provided evidence for the up-regulation of expression in relation to controls at 12DAI, as well as the down-regulation of members at both time points. RT-qPCR analysis of the expression of miRNA171a-3p-Ma06_g38750 and miRNA171e-3p-Ma06_g38750 revealed an expected negative correlation with expression of the target *Musa* gene encoding a scarecrow-like protein, supporting the involvement of this miRNA family in the biotic stress response.

MiRNA396 family members target Growth Regulating Factor (*GRF*) transcription factor genes involved in both plant development and stress responses [82]. Evidence has also been provided for a role in defence in citrus to HLB disease, targeting both proteases and LRR receptor-like kinases [83]. The reduced accumulation of miRNA393 in *Arabidopsis* has also been shown to increase PTI responses to biotrophic and necrotrophic pathogens [84]. Here, 17 members were characterized, targeting genes encoding growth relating factor 1 proteins. Tendencies were observed, although not significant, for both up- and down-regulation at both 3 and 12DAI on the basis of read abundance.

MiRNA167 members target auxin response factor (*ARF*) genes [69], with this miRNA family similarly induced by flg22 [60] and down-regulated in response to the necrotrophic pathogen *V. dahliae* in *Solanum melongena* [85]. Positive modulation has also been described in *Arabidopsis* in the compatible interaction with *Heterodera schachtii* [76], with the data here revealing 14 members expressed across the treatments and frequent greater abundance at 12DAI during the response to *P. musae,* although not significant. RT-qPCR data for miRNA167b-3p-Ma11_g01530, miRNA_167a-5p-Ma11_g01530 and miRNA_167a-3p-Ma11_g01530 pairs revealed miRNA up-regulation and repression of the target gene *ARF 12* during the response to the fungal pathogen. Similarly, RT-qPCR examination of the miR167a-3p-Ma04_g10650 pair revealed miRNA up-regulation and down-regulation of the target gene encoding an acireductone dioxygenase 1,2-dihydroxy-3-keto-5-methylthiopentene dioxygenase 1.

The miRNA family 172 has been reported to be involved in plant development in *Arabidopsis*, with gene targets including Apetala2-like transcription factors involved in flowering [71]. In the *O. sativa*–*M. oryzae* pathosystem, up-regulation has also been described in both incompatible and compatible reactions [61]. Here, from a total of 14 members, read abundance, although not significant, revealed down-regulation in two members at 3 and 12DAI, with predicted target genes encoding auxin-responsive IAA6 proteins and floral homeotic protein APETALA 2 proteins.

Members of the miRNA family 160 target auxin response factor (*ARF*) genes [86]. Not surprisingly, given the critical role of auxin in the development of feeding sites for cyst and root-knot nematodes, the members have been reported to be modulated in the *Arabidopsis*–*Heterodera schachtii* compatible interaction [76]. Whilst up-regulation in response to biotic stresses has been observed [87,88], including the PAMP, flg22 [60], miRNA160 family members have also been reported to be repressed during infection in *Pinus taeda* by *Cronartium quercuum* f. sp. *fusiforme* [75] and in *S. melongena’s* responses to *V. dahliae* [85]. With 10 members observed across the treatments, targeting expected *ARF* genes, down-regulation was observed at both time points during the incompatible interaction here with *P. musae*, with in silico data for the down-regulation of miRNA160b at 3DAI being statistically significant, indicating that fine modulation of this miRNA family involved in auxin signalling may be involved in the pathogen resistance response.

MiRNA164 family members target expression of the NAC domain encoding transcription factors [89] involved in auxin signalling and lateral root development [90]. In *Musa*, this miRNA has been shown to be up-regulated in response to temperature stress, targeting the NAC domain-containing proteins involved in development and stress responses [56]. A total of 10 members were characterized across the treatments, with evidence, although not significant, for both expression up- and down-regulation at 3DAI and 12DAI, with the miRNAs targeting a probable indole-3-acetic acid-amido synthetase GH3.5, a 14-3-3-like protein, and a probable alpha glucosidase. The RT-qPCR data for expression analysis of the latter miRNA–target gene combination (miRNA164c-5p-Ma11_g24070) revealed down-regulation of the miRNA and up-regulation of the target gene. Previously, down-regulation of this miRNA has been reported in response to *M. oryzae* [52].

MiRNA168 targets and regulates AGO1, a central enzyme in the miRNA pathway and the RNAi-induced silencing complex [91]. As such, regulation of AGO1 impacts its availability and the subsequent silencing action of additional miRNA families. Changes in miRNA168 and AGO1 expression have been observed in different plant–pathogen interactions. For example, up-regulation has been described in *Brachypodium distachyon* in response to *Magnaporthe oryzae* [92], with down-regulation occurring in *G. max* during interaction with *P. sojae* [80]. Here, a total of nine miRNA168 family members were characterized, with in silico evidence, although not significant, for down-regulation at both 3 and 12DAI. Fine-tuning of AGO1 expression and its impact on the overall RNAi machinery is likely an important component of plant–pathogen interactions [93].

Previously, miRNA 535 family members have been reported to be modulated during physiological responses to salinity stress in *Musa*, with down-regulation associated with the up-regulation of target genes encoding serine/threonine-protein kinase enzymes [67]. With eight members characterized here, in silico data, although not significant, revealed a read abundance indicative of down-regulation at 3DAI and up-regulation at 12DAI. RT-qPCR analysis revealed a stable expression of miRNA535d-5p and the target gene Ma08_g07960, encoding the 30S ribosomal protein S17.

MicroRNA159 family members, which target MYB transcription factors [87], are involved in plant development, metabolism and the response to biotic and abiotic stress [94]. The role of miRNA159 in responses to the root-knot nematode *Meloidogyne incognita* has been demonstrated in *Arabidopsis* miRNA159 mutants [95], with down-regulation reported in *T. aestivum* during interaction with *B. graminis* [43] and in *Pinus* during interaction with *C. quercuum* [75]. In *Musa*, this miRNA has been shown to be down-regulated in response to heat stress, with the target gene a PCF/TCP transcription factor associated with plant development [56]. Up-regulation, by contrast, has been reported in the interaction with *F. oxysporum* f. sp. *cubense* TR4 [58]. As negative regulators of ABA signalling pathways, these authors speculate that the family may result in the activation of SA pathways in immune responses to the fungal pathogen. Here, a total of seven members were characterized, targeting MYB transcription factors, with in silico evidence, although not significant, for the down-regulation of two members during interaction with *P. musae.*

The miRNA169 family contains four loci in *Arabidopsis*, with target host genes involved in a number of different processes related to plant defence and development [96]. Also responsive to PAMPs such as flg22 [60], down-regulation in *Z. mays* during interaction with *C. graminicola* has also been described, associated with increased JA signalling [81]. Here, a total of six members were expressed across the treatments, with non-significant in silico data revealing positive expression modulation during interaction with *P. musae* at 3DAI.

As the first plant miRNA to be shown to be modulated in response to biotic stresses [47], miRNA393 members, as observed here, can target Transport Inhibitor Response 1 (TIR1)/Auxin-signalling F-Box (AFB) auxin co-receptors [97], typically repressing auxin signalling and enabling increases in salicylic acid in response to hemibiotrophic pathogens. MiRNA393 has been proven through knockdown experiments to play an immune response role to *Phytophthora sojae* in *G. max* [80], with expression induction observed in the *Arabidopsis–P. syringae* interaction [88]. With six members characterized here, slight down-regulation, although not significant, was observed at both 3 and 12DAI with *P. musae,* and potentially indicative of an involvement in reduced SA signalling. The RT-qPCR data at 3DAI for the miRNA–target gene pair miR393b-3p-Ma06_g18000 also indicated regulation of the target gene *TIR1*, with a negative correlation of expression during the response to *P. musae* compatible with a down-regulation of the expression of auxin receptors and subsequent enhanced resistance, as also observed in responses in *Arabidopsis* to *P. syringae* [47].

### 3.3. Less Abundant miRNAs and Target Genes Potentially Involved in Plant Immune Responses

Additional, less abundant, miRNA families characterized here that are potentially involved in immune responses include miRNA398, which is known to target *ERF* genes involved in phytohormone disease resistance pathways [46,98]. Here, significant up-regulation was observed at 3DAI based on in silico data.

In the absence of pathogen challenge, the miRNA family 482 is involved in the silencing of NLR genes involved in effector-triggered immune responses in different plant species, through the production of phasiRNAs [99,100]. As such, overexpression of miRNA482 can enhance susceptibility, as observed in *V. dahliae* infection in *Solanum tuberosum* [101]. Conversely, in the presence of pathogens, miRNA482 can be silenced, with subsequent increased expression of target NLR genes resulting in disease resistance [100]. Here, one family member was characterized, with down-regulation observed at both 3 and 12DAI, although not significant, in line with NLR de-repression during pathogen challenge.

Similarly, other miRNAs involved in NLR gene repression have also been found, with miRNA2118 also producing phasiRNAs, originally identified as targeting the conserved P-loop motif in the nucleotide-binding domain in *M. truncatula* [99]. Here, although not significant, both down-regulation at 3DAI and modest up-regulation at 12DAI were observed, targeting the NLR disease resistance protein RPS2.

The plant miRNA family 397 targets laccase genes *LAC4* and *LAC17*, affecting lignin biosynthesis and cell wall thickening [102,103,104]. Additional gene targets have also included *HSP40* (*HEAT SHOCK PROTEIN 40*), *LEA* (*LATE EMBRYOGENESIS ABUNDANT*), and *SPRY* (*SPla* and *RYanodine RECEPTOR*), all of which encode proteins involved in wood formation, as described in *Populus tomentosa* [105]. Here, statistically significant in silico expression data revealed up-regulation at 3DAI, with the miRNA397a-5p targeting a pentatricopeptide repeat-containing protein Ma03_g128502. The RT-qPCR investigation of expression levels revealed a down-regulation of the miRNA in relation to the target gene.

### 3.4. Novel Musa miRNAs

A total of 24 putative novel *M. acuminata* var. Calcutta 4 miRNAs were also identified in the expression data for inoculated and non-inoculated datasets of the plant’s response to *P. musae*, representing a first description in this *Musa* genotype. Although no significant modulation of expression was observed, the diverse gene targets of these miRNAs may indicate several distinct mechanisms involved in the miRNA-mediated defence response.

## 4. Materials and Methods

### 4.1. Plant Material and Fungal Cultures

Plantlets of *M. acuminata* subsp. *burmannicoides* var. Calcutta 4 (AA) (International Transit Centre accession ITC0249) were obtained from Embrapa Cassava and Tropical Fruits. After regeneration from tissue culture, in vitro rooting was achieved by growing on Murashige and Skoog medium for a 30-day period. Plantlets were transplanted into 1.5 L pots containing sterilized substrate composed of a mixture of soil and sand (1:1), fertilizer and lime. Plant adaptation was then conducted for a period of 30 days prior to the bioassays using a growth chamber (Biochambers, Winnipeg, MB, Canada) set at a 12 h light/12 h dark photoperiod, 85% relative humidity and an average temperature of 25 °C.

*P. musae* inoculum (strain 15EB) was isolated in the field from sporodochia on symptomatic leaf lesions in *M. acuminata* Cavendish Grande Naine at the Biology Experimental Station at the University of Brasilia. Pure culture from conidiospores was obtained following transfer onto V8 solid medium with chloramphenicol and incubation at 25 °C and a 12 h light period. Total DNA was extracted from 50 mg of macerated young mycelial tissue according to the method of [106]. Genomic DNA quantification was performed via comparison with a Low DNA Mass Ladder^®^ (Invitrogen), following electrophoresis on 1% agarose gels. Molecular-based identification of the isolate was confirmed following the sequence analysis of the ribosomal DNA Internal Transcribed Spacer regions, together with actin, elongation factor1*α* and histone H3 genes [2,3,107]. Sequence identity was determined using the program BLASTn [108], against the NCBI nucleotide nr database.

### 4.2. Bioassays

The youngest fully emerged leaf from *M. acuminata* var. Calcutta 4 plants was artificially inoculated with a suspension of 1.7 × 10^4^ conidiospores mL^−1^ of *P*. *musae* and the surfactant Tween 20 at 0.05% (2.5 cm^2^ area). Non-inoculated control plants were inoculated with a water–Tween 20 (0.05%) surfactant mixture. A total of three independent replicates were collected for each sample. Leaf total RNA samples were extracted at 3 and 12 days after inoculation (DAI) for inoculated (I) and non-inoculated (NI) samples, in accordance with time point investigations by Cavalcante and colleagues (2011).

### 4.3. Musa Total RNA Extraction and Illumina Sequencing

For miRNA characterization, total RNA was extracted from liquid nitrogen-frozen leaf tissues using the Plant RNA Purification Reagent (Invitrogen, Thermo Fisher Scientific, Waltham, MA, USA), then purified using the ReliaPrep™ miRNA Cell and Tissue Miniprep System (Promega, Madison, WI, USA). RNA concentration and integrity were examined on 1% agarose gels and on a NanoDrop Lite spectrophotometer (Thermo Fisher Scientific, Waltham, MA, USA), using a quality cut-off value of 1.8 for the A260:280 ratio.

A total of 12 10 µg total RNA samples, representing the stress and control treatments, as well as biological replicates, were shipped in RNAstable™ (Biomatrica, San Diego, CA, USA) according to the manufacturer’s instructions. The RNA Integrity Number (RIN) of the resuspended RNA samples was measured using an Agilent 2100 Bioanalyzer (Agilent Technologies, Palo Alto, CA, USA). Following the NEBNEXT Multiplex Small RNA Library preparation for Illumina with size selection (New England BioLabs Inc., Beverly, MA, USA), *Musa* miRNA-containing indexed samples were sequenced in a single flow cell lane using Illumina HiSeq 2500 SR50 technology (Illumina Inc., San Diego, CA, USA) at the Génome Québec Innovation Centre, Canada. Single-read 50 bp sequencing was conducted using TruSeq RNA Chemistry v3 (Illumina Inc., San Diego, CA, USA). Illumina miRNA sequence reads were deposited in the NCBI Sequence Read Archive database under BioProject ID PRJNA937924.

### 4.4. Gene Expression in M. acuminata var. Calcutta 4 during Interaction with P. musae

For the analysis of gene expression in *M. acuminata* var. Calcutta 4 during interaction with *P. musae*, from the same 12 total RNA samples, representing the stress and control treatments, together with their biological replicates, data were accessed from Pinheiro et al., 2022 [16]. In this previous study by our group, cDNA libraries were prepared using a TruSeq RNA Library Prep Kit (Illumina, San Diego, CA, USA), and paired-end (2 × 100 bases) sequencing was conducted using TruSeq RNA Chemistry v3 (Illumina Inc., San Diego, CA, USA). All multiplexed libraries were sequenced on a single flow cell lane of an Illumina HiSeq 4000 sequencing system (Illumina Inc., San Diego, CA, USA), again, at the Génome Québec Innovation Centre, Canada.

### 4.5. Sequence Trimming and miRNA Identification

Using the stand-alone program CUTADAPT [109], raw sequence reads were quality assessed and the 3’ adapters removed. Only sequences between 20 and 24 nucleotides were preserved for subsequent analyses, with the program RFam also employed to eliminate sequences for other types of non-target RNA, such as ribosomal RNA [110].

For hairpin sequence identification and miRNA gene identification, high-quality sequences were aligned to the reference genome sequence for *M. acuminata* ssp. *malaccensis* var. DH-Pahang (https://banana-genome-hub.southgreen.fr/download), accessed on 1 March 2022 [14,111] using the program BOWTIE V.1.2.2 (http://bowtie-bio.sourceforge.net/index.shtml), accessed on 1 March 2022. Known and novel plant miRNAs were predicted from high-quality sequences using the programs Mireap (https://tools4mirs.org/software/sequencing_analysis/mireap/), accessed on 1 March 2022, and ShortStack (https://github.com/MikeAxtell/ShortStack), accessed on 1 March 2022 [112,113]. The program ShortStack provides an analysis on the basis of the criteria necessary for the identification of plant miRNAs, namely: (a) a precursor shorter than 300 nt; (b) a single miRNA:miRNA* duplex; (c) absence of secondary stems or large loops interrupting the duplex; (d) expression confirmed by sRNA-Seq; (e) result replication, i.e., present in at least two sequenced libraries; (f) precision of at least 75% in terms of the pairing of mature miRNA bases within the miRNA/miRNA* duplex; and (g) mature plant miRNA 21 nt in length. Sequences are classified as Y (yes), when all criteria are met, as N15 when all criteria are met but with the absence of miRNA*, or as N1-N14 when any criteria are absent. For manual curation, only those potential miRNAs that were classified as Y or N15 were considered.

Known miRNAs were determined by aligning the sequencing reads to published mature miRNA and miRNA precursor sequences available in miRbase V.22.1 [114], allowing zero mismatches in the first 18 nucleotides and up to two mismatches thereafter. MiRNAs were considered conserved when E-values ≤ 1 ×10^−4^ were observed and the sequences were between 20 and 22 nt in length. Reads that were not mapped to miRNAs from miRbase were considered putative novel miRNAs. Hairpin structures were analysed using Forna (force-directed RNA) [115].

### 4.6. In Silico Analysis of Differential Expression of miRNAs

Reads mapped to each particular miRNA were clustered based on their sequences and treated as isomers of mature microRNAs. The program ShortStack, in addition to analysing criteria for miRNA identification, also generates individual files of raw sequence reads for each miRNA. These files were employed for the analysis of the expression of each miRNA for each time point treatment, in comparison to the respective non-inoculated controls.

The program EdgeR [116] was employed to normalize counts according to the size of each library. Each miRNA was examined for differential expression at a particular time point treatment (3DAI_I and 12DAI_I), in comparison to the respective non-inoculated controls (3DAI_NI, 12DAI_NI). The Benjamini–Hochberg algorithm was employed to estimate the fold change in expression (FC), with miRNA DEGs considered significant when the relative expression between the inoculated and non-inoculated control treatments displayed a log_2_FC with a False Discovery Rate (FDR)-adjusted *p*-value (padj) of 0.1.

### 4.7. miRNA Target Gene Prediction

Potential host gene targets modulated by the miRNAs were predicted using TargetFinder [117]. Alignment was conducted against the RNAseq data for the *M. acuminata* var. Calcutta 4 × *P. musae* interaction time points [16]. Stringent criteria were employed to avoid false target gene prediction, with the program penalizing each alignment between an miRNA and a transcript according to the following metrics: (a) mismatches, gaps or bulges of single nucleotides: 1 point; (b) G:U base pairs: 0.5 points; (c) penalties doubled if the metrics occur between bases 2 and 13, relative to the 5′ orientation of the miRNA; (d) alignments are rejected if there is more than one gap or single nucleotide bulge, more than seven mismatch events, G:U pairings, bulges and gaps, or if there are at least four mismatches or four G:U pairings. On the basis of the TargetFinder results, up to five predicted target genes per miRNA with a score of up to four were selected, to ensure a detailed analysis of the potential targets.

### 4.8. Gene Ontology (GO) Analysis of Target Genes

Gene ontology (GO) analysis of the predicted target genes of all known and novel *M. acuminata* var. Calcutta 4 miRNAs was conducted using the program InterproScan, version 5.46–81.0 (https://www.ebi.ac.uk/interpro/search/sequence/, accessed on 20 March 2023). Molecular, biological and cellular functions of each target gene were determined, with categorization based on the functional level.

### 4.9. In Silico Expression Analysis of Target Genes

In silico gene expression data for *Musa* target genes were obtained from the previous RNAseq study conducted recently by our group on *M. acuminata* var. Calcutta 4 during interaction with *P. musae*. Data were obtained from the same 12 total RNA samples that represent stress and control treatments [16]. In this previous study, raw RNAseq sequence reads were quality assessed using FastQC Report (https://www.bioinformatics.babraham.ac.uk/projects/fastqc/), accessed on 1 June 2022. Sequence reads were selected based on quality (Fastq QC > 20) and a minimum size of 36 bp, using the program Trimmomatic [118]. High quality sequences were then aligned to the reference *M. acuminata* DH-Pahang genome sequence (available on the Banana Genome HUB platform—https://banana-genome-hub.southgreen.fr/organism/Musa/acuminata), accessed on 1 June 2022, to enable mapping to gene regions. The alignment of sequences from each library was performed using a batch mode, to allow the results to be analysed individually, using the STAR program in the twopassMode Basic mode [119]. In silico analysis of differential gene expression was conducted on aligned sequences using HTSeq-count (Python Software Foundation, Portland, OR, USA) [120]. EdgeR [121] was then employed to determine statistically significant differentially expressed genes (DEGs), based on the comparison of inoculated treatment time point datasets against equivalent non-inoculated controls. The Benjamini–Hochberg algorithm was employed to estimate fold change in gene expression (FC). DEGs were considered significant when the relative gene expression between the inoculated and non-inoculated control treatments displayed at least a two-fold FC (log_2_FC threshold > 2.0 or <−2.0), considering a False Discovery Rate (FDR)-adjusted *p*-value (padj) of 0.05.

### 4.10. cDNA Synthesis for Musa miRNAs

Expression analysis of mature miRNAs was conducted using stem-loop pulsed reverse transcription PCR according to [122] for elongation of target miRNAs. Analysis focused on early-stage interaction treatments at 3DAI to enable comparison of the in silico miRNA expression levels here with the target gene expression data published by [16]. RT master mixes for pulsed RT-PCR cDNA synthesis for miRNAs were prepared using the same RNA samples employed in Illumina sequencing (2 µg) (10 µL), and Oligo dT (500 ng/µL), together with a 10 mM dNTP mix (1 µL). Samples were heated to 65 °C for 5 min, placed on ice for 2 min, centrifuged briefly, then the following reagents were added: 5× First-Strand buffer (4 µL), 0.1 M DTT (2 µL), RNAseOut (40 units/µL) (0.1 µL), SuperScript IV RT (200 units/uL) (0.5 µL), H_2_O DEPC (0.4 µL) and a pool of stem-loop RT primers (1 µM) (1 µL), totalling 20 µL. Following RT reaction sample preparation, the samples were centrifuged briefly, loaded onto a thermocycler and incubated at 16 °C for 30 min. Pulsed RT-PCR was then conducted through 60 cycles at 30 °C for 30 s, 42 °C for 30 s, and 50 °C for 1 s. A final incubation step at 85 °C for 5 min was included to inactivate the reverse transcriptase enzyme.

### 4.11. cDNA Synthesis for Musa Target Genes

For cDNA synthesis for target genes in *M. acuminata* var. Calcutta 4, samples were prepared for each experimental condition at 3DAI from the original total RNA employed for the RNAseq and from each of the three biological replicates. Total RNA (2 µg) was reverse transcribed to cDNA using Super Script IV RT and Oligo(dT) primers (Invitrogen, Carlsbad, CA, USA).

### 4.12. RT-qPCR Expression Validation of miRNAs and Predicted Target Genes

RT-qPCR experiments were conducted according to [122]. Reactions were prepared using a Platinum SYBR Green qPCR Super Mix-UDG w/ROX kit (Invitrogen, Carlsbad, CA, USA), with amplifications performed using an ABI StepOne^®^ Real-Time PCR thermocycler (Applied Biosystems, Foster City, CA, USA). A total of three independent experimental replicates and three technical replicates per amplification were included. Ubiquitin 2 (*UBQ2*) and GTP-binding nuclear protein (*RAN*) genes were employed as references for normalization, according to [123]. All primer sequences employed are listed in Appendix A. Relative gene expression levels were quantified using the comparative ∆∆Ct method [124].

PCR reactions contained 2 µL of a 1:20 dilution of template cDNA, 0.2 µL of each forward and reverse primer, (10 mM), 5 µL Platinum^®^ SYBR^®^ Green qPCR Super Mix-UDG w/ROX kit (Invitrogen, Carlsbad, CA, USA) and adjustment to a final volume of 10 µL using nuclease-free water. Amplifications were conducted using the initial step of 52 °C for 2 min and 95 °C for 10 min, followed by a total of 40 cycles of denaturation at 95 °C for 15 s, then primer annealing and extension at 60 °C for 60 s. The specificity of primers was determined using SDS 2.2.2 software (Applied Biosystems, Foster City, CA, USA) for analysis of the Tm (dissociation) of amplified products.

Raw ΔRn data were analysed using LinRegPCR, version 2017.1. Baseline fluorescence was corrected by reconstructing the log-linear phase, with the data used to determine the RT-qPCR mean efficiency of each gene. Subsequently, the software Biogazelle qBasePlus (Biogazelle NV, Ghent, Belgium) was used to calculate the average quantification cycles (Cqs) per gene and the software GraphPad Prism v7 for statistical analysis.

## 5. Conclusions

Investigation into the miRNAs in *M. acuminata* var. Calcutta 4 and their roles in gene expression modulation during interaction with *P. musae* provides a resource for the development of efficient methods for the control of Sigatoka leaf spot disease. From a total of 226 miRNAs characterized in the study, 24 novel miRNAs were identified in this resistant genotype. Conserved miRNA members include those from families miRNA156, miRNA166, miRNA171, miRNA396, miRNA167, miRNA172, miRNA160, miRNA164, miRNA168, miRNA159, miRNA169, miRNA393, miRNA535, miRNA482, miRNA2118 and miRNA397, all of which are known to be involved in plant immune responses. A total of 76 gene transcripts were targeted during infection. GO classification indicated numerous molecular activity terms potentially related to defence responses that included transcription factor activity, nucleic acid binding, nucleotide binding, oxidoreductase activity and protein kinase activity, together with Biological process terms such as regulation of transcription, response to hormone, oxidation–reduction, and response to oxidative stress. When considering the phylogenetically close relationship of *P. musae* with *P. fijiensis* and *P. eumusae*, together with the observed expression of certain miRNA families in both *Musa*–*P. musae* and *Musa*–*F. oxysporum* f. sp. *cubense* interactions, an examination of the potentially conserved roles of certain *Musa* miRNAs to diverse pathogens is warranted. The characterization of miRNAs and their target genes in the host provides a basis for future genetic manipulation through RNA interference-based strategies for the development of stress-resistant cultivars. These strategies can include, amongst others, short hairpin RNA-mediated gene silencing, the development of artificial miRNAs, MIR gene knock-out and the nano particle delivery of miRNAs. Such strategies offer potential for enabling resistance against diverse stresses across numerous crop species [28,125]. The characterization of miRNAs involved in cross-kingdom transfer and their modes of action may also advance our understanding of the complex interactions between plants and pathogens, with potential manipulation for reducing pathogen virulence or increasing plant immune responses likely to benefit plant productivity. In this context, ongoing high-throughput sequencing and bioinformatic tool development by our group will enable the prediction of miRNAs, milRNAs, and gene targets in both the plant host *M. acuminata* var. Calcutta 4 and the fungal pathogen *P. musae*.

## Figures and Tables

**Figure 1 plants-12-01473-f001:**
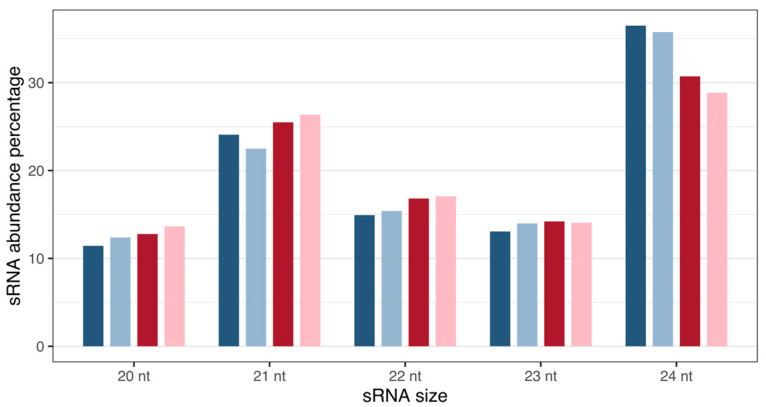
Size distribution and percentage abundance of *M. acuminata* var. Calcutta 4 small RNA sequences. Blues: 3DAI. Reds: 12DAI. Dark-coloured bars: non-inoculated samples; light-coloured bars: inoculated samples. Abbreviation: nt, nucleotides.

**Figure 2 plants-12-01473-f002:**
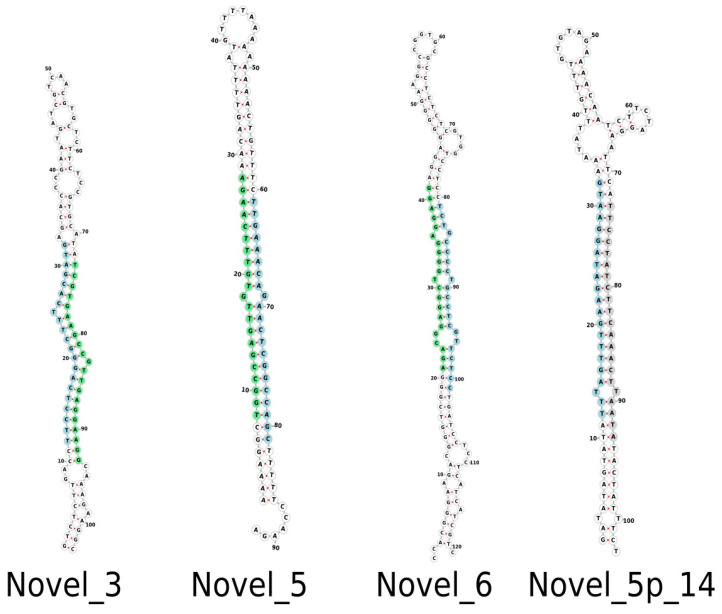
Examples of predicted hairpin secondary structures in selected novel *Musa acuminata* var. Calcutta 4 miRNAs. Secondary structures were determined using the program Forna. Mature sequences are highlighted in light blue. Star sequences are highlighted in light green. A putative star sequence is highlighted in light grey.

**Figure 3 plants-12-01473-f003:**
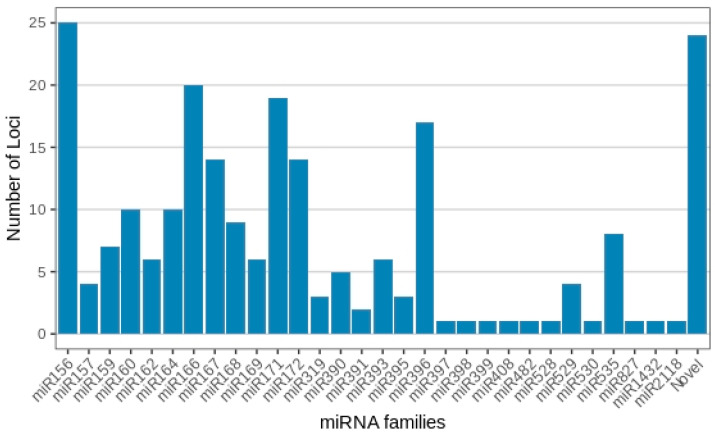
*Musa acuminata* var. Calcutta 4 miRNA families and member numbers observed in the small RNA libraries representing the four experimental treatments.

**Figure 4 plants-12-01473-f004:**
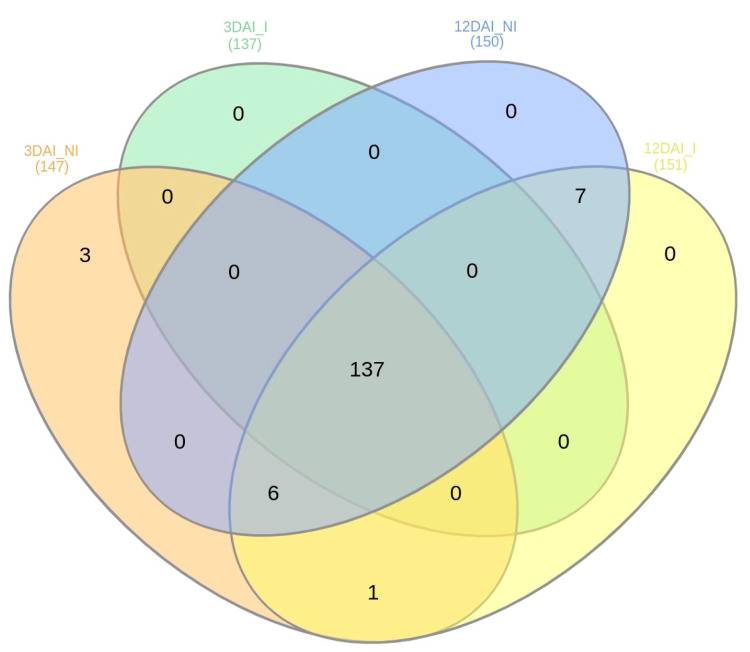
Numbers of abundant *Musa acuminata* var. Calcutta 4 miRNAs observed across the four small RNA libraries, each with an excess of 1000 read counts.

**Figure 5 plants-12-01473-f005:**
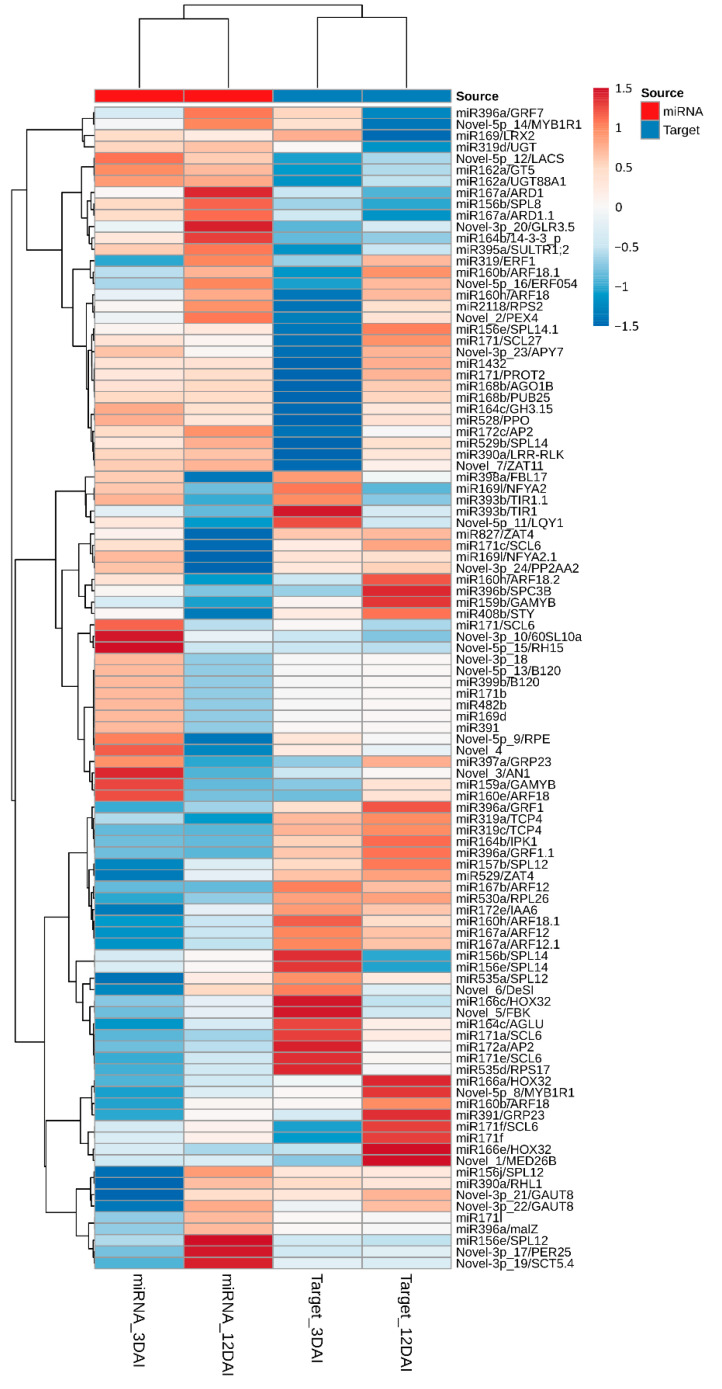
Hierarchical clustering heatmap of differentially expressed *Musa acuminata* var. Calcutta 4 miRNAs and target genes. Log2Fold-based changes in gene expression represent *Pseudocercospora musae*-inoculated treatments in relation to non-inoculated controls at each time point (3 and 12 days after inoculation). Original in silico expression values for the miRNAs and target genes were normalized with EdgeR. Changes in expression were depicted from high (red) to low (blue), according to the colour scale. Abbreviation: DAI, days after inoculation.

**Figure 6 plants-12-01473-f006:**
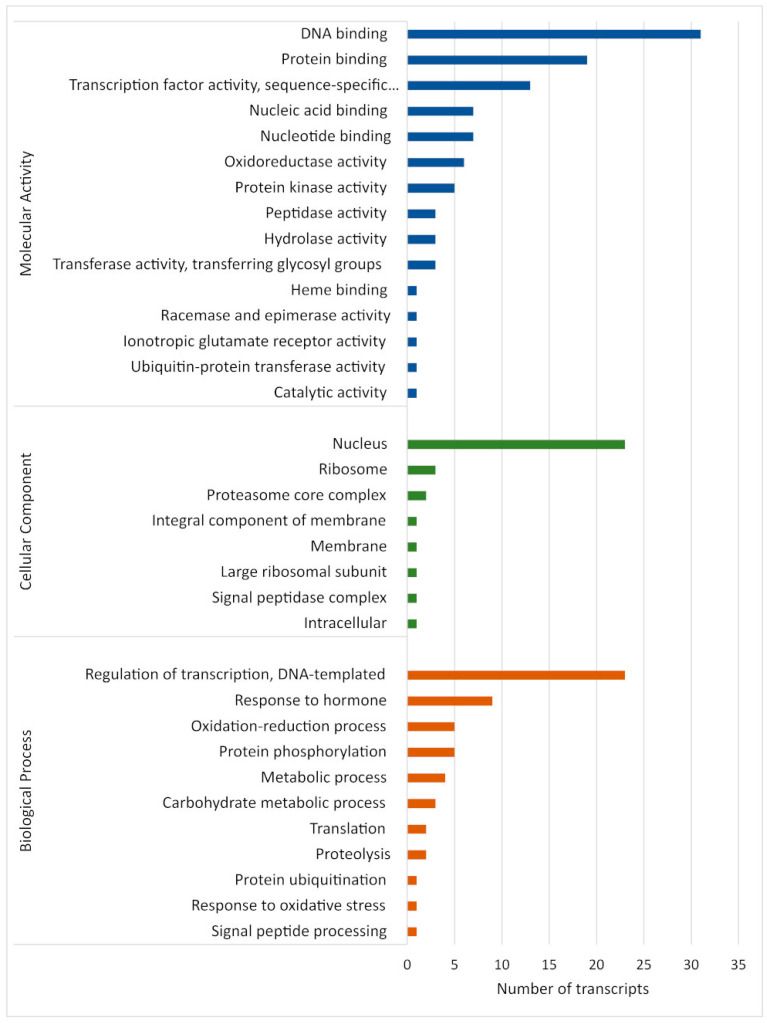
Distribution of enriched gene ontology (GO) terms for gene targets of *Musa acuminata* var. Calcutta 4 miRNAs.

**Figure 7 plants-12-01473-f007:**
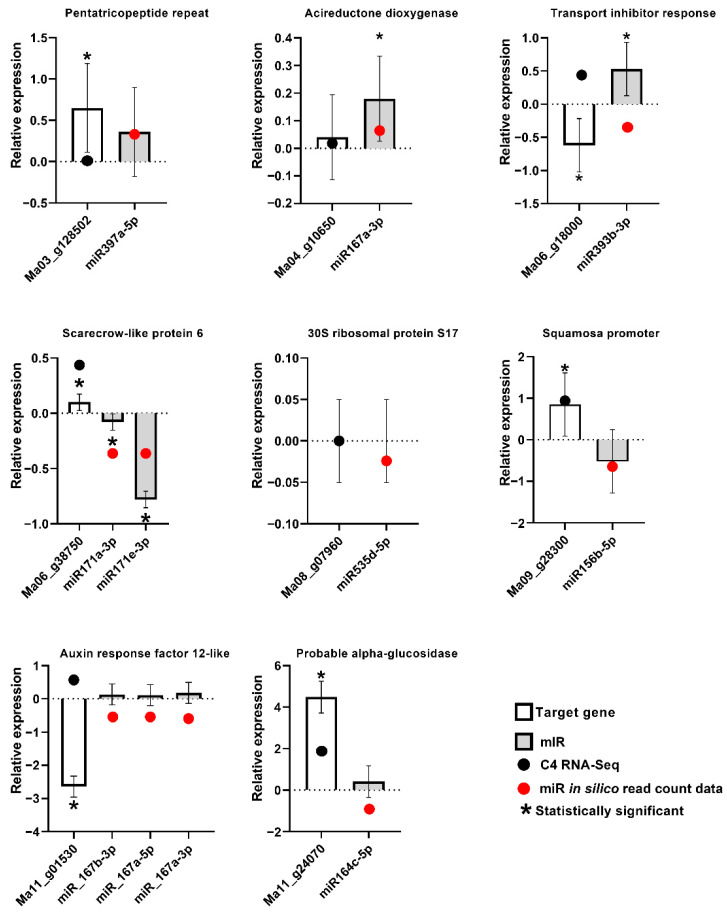
Stem-loop RT-qPCR and conventional RT-qPCR validation of differential expression profiles for selected miRNAs and their target genes in *Musa acuminata* var. Calcutta 4. Standard errors were calculated based on data for three biological replicates per treatment and three technical RT-qPCR amplification replicates.

## Data Availability

The data that support the reported results can be found at the NCBI Sequence Read Archive (SRA) database (BioProject ID PRJNA937924).

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
