# Peer review of "Characterization of microRNAs and Target Genes in Musa acuminata subsp. burmannicoides, var. Calcutta 4 during Interaction with Pseudocercospora musae"

_plants, 2023, doi:10.3390/plants12071473_

Round 1

Reviewer 1 Report

This article presents a characterisation of microRNAs and their target genes in banana during interactions with the fungus Pseudocercospora musae. The results are novel, done in the state of the art, well presented and thus deserve to be published. I have only a few suggestions to improve a little bit the text:

L51: This fungus belongs to the species complex called Sigatoka leaf spot complex. In particular the species Pseudocercospora fijiensis of this complex is nowadays predominant in the world. The authors worked with the species P. musae certainly because it was the only one present in Brazil until recently. However, the species P. fijiensis is still spreading. I think it is important to introduce this information in order to give perspectives on P. fijiensis in the discussion/conclusion.

L131: As the M&M is at the end of the article it will be useful to present here a summary of the overall approach taken.

L294: The term 'family' is used for the first time here and has not been presented before. How are these families defined?

L346: 'with previous findings in Musa'. Which context?

L755: I will propose somewhere in the conclusion some perspectives for comparison with other pathogen of banana including, for foliar disease, the more pathogenic and now predominant P. fijiensis species. Can we consider as generic the microRNAs that have been demonstrated in banana interaction with P. musae and F. oxysporum fsp cubense, which are nevertheless very different pathogens? Should research on such microRNAs be pursued as a priority?

Reviewer 2 Report

1. Many grammatical errors have been found in the manuscript. So, here is a need of

    deep revision of every line.

2. Written English is generally very poor but, perhaps, doesn't interfere with meaning as

    long as the reader is prepared to put in the effort.

3. The quality of Figure 1 and 5 should be improved.

4.  Line no. 68-69 should be rewritten in the manuscript.

5. Introduction is too lengthy. Only key points should be mentioned in the manuscript.  

6.  Repetition should be avoided throughout the manuscript. For instance, line no. 75-76.

7. Author should maintain a uniformity in writing. Write down either the botanical or

   common name of plants in the entire manuscript.8

8. Paragraphs are repeated in the section “Results: Identifications of miRNAs”. Therefore, the

    repeated paragraphs should be removed.

 9. Results of section “Differential gene expression heatmap representations of miRNAs

    and target genes” should be rewritten.

10. Discussion is too bulky. Make the discussion simple and concise. Author should discuss only key points.      

Round 2

Reviewer 2 Report

Accepted